# Barriers to contraceptive use in humanitarian settings: Experiences of South Sudanese refugee women living in Adjumani district, Uganda; an exploratory qualitative study

Roselline Achola[1]*, Lynn Atuyambe[1], Elizabeth Nabiwemba[1], Mathew Nyashanu[2], Christopher Garimoi Orach[1]

1 Department of Community Health and Behavioural Sciences, College of Health Sciences, Makerere University, Kampala, Uganda, 2 Department of Health & Allied Professions, School of Social Sciences, Nottingham Trent University, Nottingham, United Kingdom

* rosellineac@gmail.com, rachola@musph.ac.org

## Abstract

### Introduction

Contraceptive use can be lifesaving, empowering and cost-effective for women and girls. Access to contraception is still challenging to female refugees due to several barriers including language, low educational level, lack of information, influence by family members, limited income, cultural and religious norms. This study explored barriers to contraceptive use among South Sudanese refugee women living in Adjumani district, Uganda.

### Methods

An exploratory study design using qualitative methods were employed involving women of reproductive age (15–49 years). Purposive sampling was used to select participants for Focus Group Discussions (FGDs) and In-depth Interviews (IDIs) from three settlements in Adjumani district. We conducted four FGDs, each consisting of 8 participants. We also conducted fourteen in-depth interviews (IDIs) with women of reproductive age. The IDI and FGD guides were translated into local languages before they were used to collect data. The interviews were recorded, transcribed verbatim and translated into English. Audio recordings were labeled before being translated back to English. Deductive, team-based coding was implemented, and a codebook developed. Transcripts were entered, and data coded using Atlas ti version 14. Data were analyzed using content analysis to produce the final outputs for the study.

### Results

The study found several challenges to contraceptive use. These included gender dynamics, socially constructed myths on contraceptive use, cultural norms, limited knowledge about contraceptives, men's negative attitudes, antagonism of contraceptive use by leaders and reprisal of women who use contraception.

**Data Availability Statement:** Yes - all data are fully available without restriction; All relevant data are within the paper.

**Funding:** Grant ID: NICHE -UGA-288. This research was conducted with financial support from Nuffic grant through The International Institute of Social Studies Erasmus University, Rotterdam for Strengthening Education and Training Capacity in Sexual Reproductive Health & Rights in Uganda (SET-SRHR). The funders had to role in the study design, data collection and analysis, decision to publish or preparation of the manuscript.

**Competing interests:** The authors have declared that no competing interests exist.

## Conclusion

The study concluded that there is need for community strategies to break down the barriers to contraception utilization among refugee women. Such strategies should involve men and women alongside gatekeepers to enhance sustainability.

## Introduction

The last twenty-five years have witnessed massive displacement of people due to political upheavals and natural disasters, violence and other forms of human rights abuses [1, 2]. This has caused the establishment of refugee settlements in low and middle income countries (LMICs) whose infrastructure and capacity for support is poor [3]. When immigrants take refuge in other countries, demand for education, health services, infrastructure, natural resources, food, land and security are considered key with lesser prioritization to family planning (FP) interventions [4]. Yet, refugee populations are affected by an array of problems ranging from economic, socio-cultural and health issues including ability to make decision for contraceptive use in the new environment [1, 2]. Contraception is one of the needs for refugee communities with limited prioritization and restrictions by some host governments on its access [5, 6]. Contraception is a key pillar to improved quality of life for refugee women considering that family size has a direct impact on family well-being and support [3, 4].

Since the start of the civil war in South Sudan in 2013, many people took refuge in Uganda. Uganda is the 3rd largest refugee-hosting country in the world and currently hosts about 1,595,405 refugees from neighbouring countries and beyond [7]. These refugees are spread all over the country with a high concentration in the west Nile region. Over ninety-four percent (1,499,681) are living in settlements established in 12 districts of Uganda and 6% (95,724) are living in Kampala City. Of these, 60.5% (965,220) come from South Sudan and 29.3%(467,454) come from the Democratic Republic of Congo and others (10.2%, n = 162,731) from Somalia, Burundi, Rwanda, Eritrea, Ethiopia and Sudan [7]. Of the total number of refugees who come from South Sudan, about 214,453 live in settlements in Adjumani district.

The refugee women have faced several challenges related to access to contraception [8]. These challenges include language barrier, lack of information, peer influence, limited income, desire to replace lost family members, side effects, socio-cultural preference and unacceptability of contraceptive use, resulting into unintended pregnancies [4, 7]. Yet, according to the Global standard for Sexual Reproductive Health response in acute emergencies, prevention of unintended pregnancies is one of the six objectives under Minimum Initial Services Package for Sexual Reproductive Health programming [9]. Family planning, according to World Health Organisation report (2020), was defined as a voluntary and informed decision by an individual or couple on the number of children to have and when to have them [9]. It is characterized by the use of contraceptive methods that include: condoms, injection, oral pills, implants, Intra-Uterine Devices (IUD) of both hormonal and Copper T, female and male sterilization, Lactation Amenorrhea Method (LAM) among others [7]. Proper use of these methods has proven effective in preventing unintended pregnancies. This would enable a mother to offer appropriate care to the baby hence increase chances of child survival [10].

Over 80% of the refugees are women and children who need life-saving, empowering and cost effective interventions including protection during humanitarian crisis [11–13]. The limited access to services makes them more susceptible to unintended pregnancies [14–16]. South Sudan is one of the countries whose modern contraceptive prevalence rate is very low at 2.7% with unmet need for FP at 30.8% and maternal mortality ratio of 789/100,000 live births [17].

In South Sudan, families live in a patriarchal society where women's position is subordinate to men who are the main decision makers including determining the number of children to be born [18, 19]. Thus, women are expected to produce as many children as possible in return for the cows paid as dowry to their parents. The women who fail to fulfill these expectations and use contraception face serious consequences [14, 20]. The consequences include intimate partner violence, separation, or divorce [21]. This study aimed at exploring the barriers to contraceptive use in humanitarian settings: challenges and opportunities for South Sudanese refugee women living in Adjumani district, Uganda.

## Methods

### Study design

An exploratory design using qualitative methods was done to explore the barriers to contraceptive use in humanitarian settings for South Sudanese refugee women living in the settlements in Adjumani district, Uganda. This method is important for deeper understanding of the problem as opposed to finding a final solution to the problem [22]. This study is guided by the ecological perspective theory which describes how different levels interact differently to cause an outcome which in this case is contraceptive use among refugees [23]. *"An ecological perspective on health, emphasizes both individual and contextual systems and the interdependent relations between the two"* [23]. We adopted this theory because decision making on health behavior is central to this study while behavior is affected by several levels of influence indicated in the ecological theory (See Fig 1 below). Ecological perspective theory describes how behavior shapes and is also shaped by the environment in which individuals live [23]. It describes eight theories and models that explain individual, interpersonal and community behavior and offers approaches to solving problems [24, 25]. The levels of influence include: 1) intrapersonal/individual factors; 2) interpersonal factors; 3) community factors including institutional/organizational factors and public policy frameworks. These theories and models helped to explain how the different levels of influence interact towards population health, which in this case is the use of contraception for improved health (see Fig 1). The interview schedule was translated into the local languages to facilitate understanding of the questions by the research participants. This study was conducted in three settlements of Pagirinya, Nyumanzi and Mirieyi in Adjumani district.

### Study population and recruitment

The participants were women of reproductive age (15–49 years), both users and non-users of contraceptives. They mainly spoke Madi and Arabic which were widely understood by most tribes in the settlements. A purposeful sampling approach [26] was used to select participants for FGDs and IDIs with support of community leaders and service providers from the health facilities. Overall, four FGDs were conducted. Each focus group consisted of 8 participants. We also conducted fourteen IDIs.

Inclusion criteria was for only females of 15–49 years and registered refugees from South Sudan.

Exclusion criteria comprised of participants who were very ill and were not residing in any of the settlements of study. Overall, 46 female participants were recruited in this study.

### Pretest

Whereas the research assistants were knowledgeable, and familiar with the study area, a tailored training was conducted on the basics of qualitative research methods and review of the

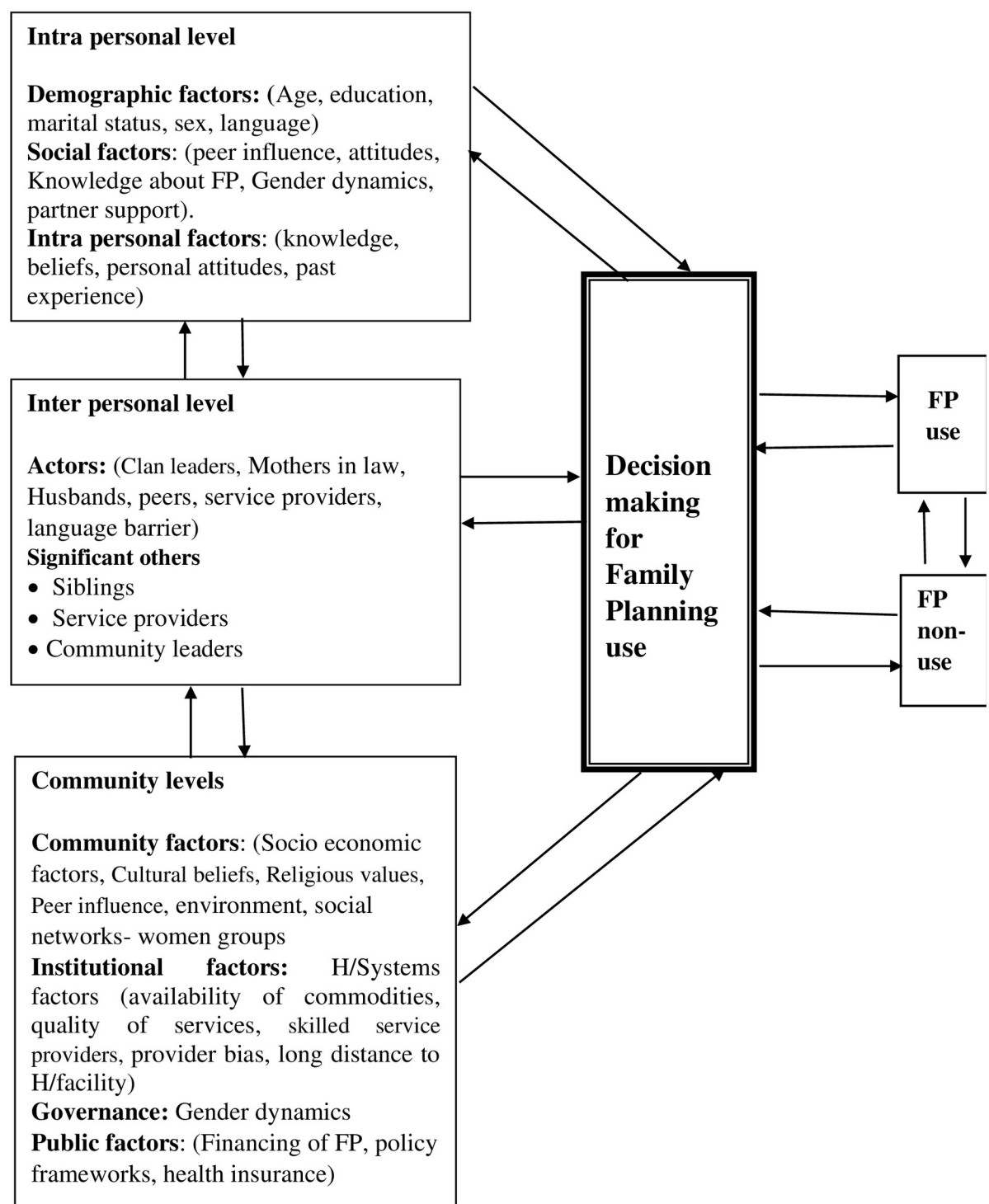

**Fig 1. Different interactions between different levels to cause an outcome of decision making for family planning use.** Adopted from Ecological perspective.

tools. Pre—testing of the tools was done in Agojo settlement in the same district. The research participants were asked about the clarity of the questions which they confirmed to be clear, thus the researchers adopted the interview guides without much alteration.

### Data collection

The trained Research Assistants (RAs) who helped to collect the data were both males and females between the ages of 24–49 years. They were identified from a pool of experienced persons from the district and Makerere University School of Public Health. Some of the RAs were social scientists with master's degrees and were able to speak the languages (Madi and Arabic) best understood by most of the refugees in the settlements and the host communities. Data were collected using translated and pre tested In-depth interview guides and focus group discussion guides that were developed based on previous studies and reviewed in relation to decision to use contraceptives [27]. The interviews were recorded verbatim, transcribed, and were translated back to English.

### Data analysis

Audio recordings from the FGDs and IDIs were labelled, transcribed verbatim, translated into English by experienced research assistants who were fluent in Madi, Dinka and Arabic as the main languages and the recordings stored. Deductive, team-based coding was done, and a codebook developed. Transcripts were entered, and data coded using Atlas ti version 14. Data were analyzed using content analysis. To ensure reliability, coding was done by three research experts led by the Principal Investigator. They read the transcripts, discussed emerging issues, and agreed on common themes. The team selected two transcripts, each one read through, assigned different meaning to each response and developed a codebook following Braun and Clarke procedure [28]. On completion, they met via zoom and discussed the descriptive codes to come up with a unified code book. While coding, there were interactive discussions arising from the discrepancies and disagreements that were resolved by having same meaning of the codes before moving to the next step.

## Results

This study presents the findings from the experiences of the refugee women from South Sudan in accessing contraceptive information and services while living in Adjumani district. We found that several challenges affected contraceptive use. These included gender dynamics, socially constructed misconceptions, cultural norms, limited knowledge about contraception, men's negative attitudes, the antagonism of community leaders on contraception and the reprisal of women following contraceptive use without permission from their spouses.

### Socio-demographic characteristics of the participants

A total of 46 refugee women participated in this study. Thirty-two participated in four FGDs that consisted of 8 participants each and 14 women participated in the IDIs. Twenty participants were young people of ages 15–19 years, fourteen women were between the ages of 20 and 30 years, ten women were between the ages of 31 to 40 years and only 2 were between 41 and 49 years. Only three participants from the young population (15–19 years) were students. Most of the women (47.8%) had no formal education. However, only 2.2% had attained tertiary education. Over 73% were housewives and 6.5% still students which explains the low economic status with only 19.6% engaged in some business. Majority were from the Catholic faith

**Table 1. Socio—demographic characteristics of the participants.**

| Variables | | Number (%) | Number (%) | Total, (%), N = 46 |
|---|---|---|---|---|
| **Data collection methods** | | FGDs | IDIs | |
| **Total number** | | **32 (100)** | **14 (100)** | **46 (100)** |
| **Gender** | | | | |
| Females only | | 32 (100) | 14 (100) | 46 (100) |
| **Age** | | | | |
| **Age range**: | 15–19 years | 16 (50.0) | 4 (28.6) | 20 (43.5) |
| | 20–30 years | 8 (25.0) | 6 (42.8) | 14 (30.4) |
| | 31–40 years | 8 (25.0) | 2 (14.3) | 10 (21.7) |
| | 41–49 years | 0 | 2 (14.3) | 2 (4.4) |
| **Tribe:** | Madi | 10 (31.0) | 4 (28.6) | 14 (30.4) |
| | Dinka | 10 (31.2) | 3 (21.4) | 13 (28.3) |
| | Nuer | 2 (6.3) | 1 (7.1) | 3 (6.5) |
| | Kuku | 2 (6.3) | 1 (7.1) | 3 (6.5) |
| | Others | 8 (25.0) | 5 (35.8) | 13 (28.3) |
| **Marital status:** | Married | 21(65.6) | 10 (71.4) | 31 (67.4) |
| | Single | 8 (25.0) | 3 (21.4) | 11 (23.9) |
| | Separated | 3 (9.4) | 1 (7.2) | 4 (8.7) |
| **Education:** | No education | 16 (50.0) | 6 (42.8) | 22 (47.8) |
| | Primary | 10 (31.3) | 4 (28.6) | 14 (30.4) |
| | Secondary | 5 (15.6) | 4 (28.6) | 9 (19.6) |
| | Tertiary | 1 (3.1) | 0 | 1 (2.2) |
| **Occupation:** | Housewife | 24 (75.0) | 10 (71.4) | 34 (73.9) |
| | Business | 6 (18.8) | 3 (21.4) | 9 (19.6) |
| | Student | 2 (6.2) | 1 (7.2) | 3 (6.5) |
| **Religion**: | Catholic | 20 (62.5) | 8 (57.2) | 28 (60.9) |
| | Anglican | 10 (31.3) | 3 (21.4) | 13 (28.3) |
| | Muslim | 1 (3.1) | 0 | 1 (2.1) |
| Others (Born again, SDA) | | 1 (3.1) | 3 (21.4) | 4 (8.7) |

(68%) and married (45.6%), (see Table 1 below), The socio—demographic characteristics of the participants.

This study was guided by the conceptual framework (ecological model) that was used to align the results which were classified under the different levels of this conceptual framework (see Fig 1 of the conceptual framework).

These include Intrapersonal, interpersonal and community levels.

## 1. The intrapersonal level

This study found that low educational level (48%) was a big factor in contraceptive use among the refugee women in Adjumani. Similarly, the study found that majority of the women (73%) were full time housewives and not engaged in any income generating activities. Given the cultural norms of marrying girls early, this study found that 43.5% of the participants were young women of ages 15–19 years. Of these 85% were already mothers.

**1.1 Limited knowledge about contraceptive methods.** The participants reported that there was limited information about contraception among refugees. They acknowledged that there was lack of health promotion messages that promoted contraception back home in South Sudan.

"...I don't use contraceptives because I don't know more about it. No one ever came to talk to us in the settlements here about its use..."

(FGD_Women_20+_Adjumani_Mirieyi _settlement_ Dinka only).

Additionally,

"...things would be different because here in Adjumani, one can mention anything about contraception which is not the same case back home in South Sudan..."

(Female FGD_20+_Adjumani_Pagirinya_ settlement).

## 2. Interpersonal level

**2.1 Gender dynamics.** *2.1.1 Patriarchal dominance of men.* The participants reported the influence of men in making decision on contraceptive use. They further reported that women find it hard to make decision to start on a contraceptive method without their husband's involvement. Majority of women reported accessing contraceptives stealthily. They acknowledged that men were the custodians of decision making in their communities. Thus, very crucial in determining family size and when to use contraceptives.

"...Because my husband was aware that my mother made decision for me to start on contraceptives without his consent, he said that if I fall sick in his house, I should go to my mother who should take care of me moving forward..."

(IDI _Female 20+_ Adjumani_Pagirinya_Settlement).

And,

"...I witnessed a sad moment when my neighbor made her decision to go to hospital and was given this method of inserting on the arm (Implant). When this woman got sick, while the husband tried to support her, he felt the implant under her arm. He immediately rushed to hospital with her and demanded that the implant be removed from the wife's arm in his presence..."

(IDI _Female_20+_Adjumani_Pagirinya_Settlement).

*2.1.2 Limited women empowerment.* The study participants acknowledged that less empowered women were not able to convince their husbands about the intention to use contraceptives. They reported that making decision to use contraception was only possible for women who are economically empowered and can sustain themselves and provide their own basic needs. This is because the husbands withdraw all the support if they discover their wives using any contraceptive methods. The women who were discovered by their husbands using any contraceptive method were denied basic support which included food, shelter, clothes, soap and medical care.

"...Because of my small business, I decided to use contraceptives knowing that if he gets me, I will support myself and my children because that will automatically result in neglect and another wife added on me..."

(IDI _Female 20+_ Adjumani_Pagirinya Settlement).

And,

"...*if you are very weak the husband will chase you away from that home...*"

(IDI_ female_ 20+_Pagirinya_Adjumani_settlement).

*2.1.3 Mistaking contraceptive use for promiscuity.* The women and girls who participated in this study reported that many husbands and parents of young women believed that women who use contraception are promiscuous. Furthermore, they alleged that the communities view contraceptive use as a cover up for promiscuity, thus strongly oppose its use by both women and girls. They further asserted that it is even worse for unmarried adolescents because their parents fear that they would become prostitutes and no man would marry them.

"...*yes, it is now more of a culture in the Dinka tribe, that women who are using contraception when the community gets to know, they will be called prostitutes because they don't want to produce more children. This has discouraged more women from using contraceptives even when they have genuine reasons...*"

(FGD- Female 20_Adjumani_ Pagirinya _settlement).

And,

"...*Most parents don't want their young girls to use contraceptives, saying the girls want to become prostitutes by covering up with prevention of pregnancy...*"

(FGD- women_ 15–19_Mirieyi settlement).

*2.1.4 Joint decision by couple against contraceptive use.* The research participants in the refugee settings viewed contraceptive use as a barrier to getting the many cows in form of bride price that comes with a girl child. Families that produced many girls are assured of big gains in form of dowry. Therefore, participants reaffirmed that some couples do agree jointly not to use contraception and have as many girls as possible to reap the benefits of dowry.

"...*in our communities couples agree jointly to shun contraception so as to produce many girls to benefit from dowry as a source of wealth...*"

(IDI_Female_20+_Adjumani_Nyumanzi_ settlement).

And,

"...*Back home in South Sudan, we even don't use contraceptives because what we want is to give birth to many children. This is the reason why we don't want to know more about your contraceptives. In our tribes, we face poverty due to lack of dowry, so we need more girls to get out of poverty...*"

(FGD_female 20+ _Adjumani_Mirieyi_settlement).

**2.2 Men's negative attitudes about contraceptives.** The research participants reported that their main barrier to using contraception is the negative attitudes of men about it. They added that men think that their women will stop producing and yet they want more children.

*"...you see these men of ours don't want contraceptives, they say it will damage people and its dangerous for women to use. Even though you share with them the benefits, they will disagree and disagree and start quarrelling and fighting..."*

(IDI _Female20+_Adumani_Pagirinya_Settlement).

Surprisingly, the women added that this negative attitude by men have also influenced the way other women perceive contraception. They reported that women also speak negatively about the use of contraception amongst their peers leading to non-use.

*"...But for women who listen to their husbands have also developed negative attitudes about contraception. They discourage other women, saying that it is not good because women may give birth to children who are dormant (with mental retardation), lame (with deformity) leading to family breakage ..."*

(IDI _Female_20+_Adjumani_ Pagirinya_Settlement).

**2.3 The reprisal following contraceptive use.** The participants reported that some women received reprisals when found to have made decision to use contraceptives despite refusal by their husbands or other family members. They asserted that it was safer when the husbands and family members remained ignorant about their contraceptive decision.

*"... Some women are neglected by their husbands without any social support. This will also cause hatred between wife and husband in the homes causing unnecessary tension and worse of all, separation where the husband chases away the woman from his home due to anger..."*

(FGD _Female 20+_Adjumani_Nyumanzi_Settlement).

Additionally,

*"...The worst thing of all is that many women have been separated from their marital homes by force as their husbands chased them away when discovered with contraceptives..."*

(FGD _Female 20+_Adjumani_Mirieyi_Settlement).

## 3. Community level

The community factors included socially constructed myths and misconceptions about contraception, antagonism against contraception and cultural and religious norms related to contraception. There were also institutional factors that affected contraceptive use among South Sudanese refugee women.

**3.1 Community socially constructed myths/misconceptions about contraceptive use.**
Most participants reported that in their communities, people have negative perceptions about contraception. They reported that contraceptives cause cancer, infertility, fibroids, burn the ovaries, damage the female eggs, and even lead to giving birth to deformed children. They further claimed that most children born with deformities in their communities is because of the effect of some contraceptive methods. In addition, some of these myths are exaggerated by

parents who use it as a threat to their daughters while discouraging them from using contraception.

"... *I think if women are not determined due to the myths that go around about various contraceptive methods, no one would probably use them given the negative forces that come your way. People say that Depo causes cancer, fibroids among others*..."

(IDI_Female 20+_Mirieyi_settlement).

Additionally,

"...*Some parents say that adolescent girls may have babies with missing body parts. Any abnormality is associated with use of contraception. Even those who don't produce after marriage are labelled as having used* contraceptives *at early age. therefore, contraceptive use among young people is very difficult in the settlements because all eyes are on them*..."

(IDI_ Female 15–19_Nyumanzi Settlement).

**3.2 Antagonism against contraceptive use by community leaders.** Many women who participated in this research reported that, there was a deliberate effort by the community leaders to antagonize family planning programs within the settlements. They continuously advocated for many children to build a stronger and bigger society, claiming that conflict back home disentangled their families that must be rebuilt with a new generation. The participants added that their leaders have a fear of losing the entire generation if contraception is embraced. This is the reason why they campaign against contraception programs for refugees. They also reported that children are viewed as source of prestige and security, thus the need for more children. They affirmed that this is a matter for clan heads and not for the individual women.

"...*it's common in Madi culture or any other culture among us the refugees that we are interested in replacing our lost people. Therefore, we need to give birth to many children who will be going to occupy our land back in South Sudan. Even if you go back there, that land is now open and vacant with no people. Thy are just advising women to produce as many children as they can*..."

(IDI female_ 20+_ Adjumani_Pagirinya_Settlement).

And,

"...*A South Sudan man who lived with a woman who had given birth to less than five children and was not conceiving again, the clan leaders forced the man to marry another wife because more children were needed*..."

(IDI female 20 and above Pagirinya settlement).

**3.3 Cultural and religious norms related to contraceptive use.** Most of the participants reported that women are married to have as many children as they can until when they naturally reach menopause. Thus, failure to have the desired number of children by the family may

lead to disciplinary action against the woman because it contravenes the cultural beliefs and norms on big family size.

*It is their cultural belief that husbands would tell their wives that I brought you to my home here to produce children and you cannot stop before your old age…"*

(IDI_female_20+_Adjumani_Pagirinya_settlement).

And,

"*…Due to Christian religion, particularly the Seventh Day Adventist, we are always told to go for advice. When you go for the so- called advice, one person will call another, then another one will call another to deal with you from all corners. So, what they will tell you is not to take any contraceptives. They will not tell you anything positive about it…"*

(*FGD Female 15–19 Pagirinya Settlement*).

**3.4 Institutional/organizational factors.**   Most participants reported various institutional factors that hindered them from making decision to use contraception. These barriers include unclear policy and legal frameworks on contraception for younger adolescents, limited options of contraceptive methods and poor-quality services.

*3.4.1 Unclear Policy and legal frameworks.* The participants from the younger group (15–19) reported that service providers denied them contraceptive services claiming that the policy on Sexual and Reproductive Health does not allow them to offer contraceptive services to adolescent girls.

"*…My friend went to Nyumanzi Health Centre and the nurse told her that she is not allowed to use contraceptives because she is still young…"*

(FGD- women_ 15–19_Nyumanzi_ settlement).

*3.4.2 Poor quality services and limited contraceptive options.* The research participants reported that there is limited choices of contraceptive methods. They added that poor quality services also existed in the different health facilities in the settlements which discouraged them to make decision to use contraception. They cited the fact that many women who received implants as Long Acting and Reversible Contraceptives (LARCS) were frustrated at the time of removal due to lack of competencies by the service provider.

"*…They tell you that contraceptives will go with your blood (become anaemic) and the health workers cannot even screen you to ascertain the actual method that is appropriate for one's blood group. This has caused fear among women and they don't go for contraceptives…"*

(IDI_ Female 20+_Nyumanzi Settlement).

Similarly, some women reported that health workers do not follow the right procedures of screening women for eligibility criteria to establish appropriate contraceptive method to offer. Additionally, the counselling services are insufficient because clients are not informed about some contraceptive methods and the possible side effects. The women also expect the health workers to take blood samples for testing before initiating them on any method of

contraception. Therefore, women have fears that these methods may lead to loss of blood through bleeding, and as such discontinue the use of contraception prematurely.

"...*The health workers don't remove the family planning method in the body when you want to early because they say it is for five years, for four years, for three years...*"

(IDI_ Females 15–19_Nyumanzi Settlement).

## Discussion

In this section, the findings were discussed based on the ecological model involving three levels. These include intrapersonal, interpersonal and community levels including institutional/ organizational level. Several factors were found to affect contraceptive use among the South Sudanese refugee women living in the settlements in Adjumani district. The study used the Ecological model to explain the complex relationship between the various factors that affected contraceptive use by women and girls in Adjumani district.

At intra-personal level, the demographic, socio- economic, cultural, religious, environmental factors, personal attitudes, language, husband support and knowledge about contraception affected internal decision-making and choice for contraceptive use. This study found that low educational level was a major factor affecting contraceptive use.

Similarly, the study found that majority of the women were full time housewives and not engaged in any socio-economic activities. This status further exacerbated their vulnerability thus, agreed that male authorization for contraceptive use is crucial because of the fear of being neglected. As such, the power relations that existed among women and men living in humanitarian settings posed imbalances that negatively impacted on contraceptive use.

The concept of "empowerment" has been central to efforts to address gendered inequalities and power imbalances, thereby accelerating development and improving the health and well-being of women and girls [29, 30]. Empowerment is framed as a process of enhancing an individual's capacity to exercise choice, make decisions, and the ability to act on those decisions, and achieving their choice. Interventions that aim to "empower" women and girls are designed to increase self-efficacy and agency, and therefore improve sexual and reproductive decisions and related health outcomes, including the decision to use contraceptives or not. As this study demonstrates; since socio-cultural gendered norms and expectations are key obstacles to women's exercise of agency in choosing to use contraceptives or not [31], as pervasive gendered sexual norms tend to give men greater decision-making power in sexual and reproductive matters. It is critical that interventions and programmes that aim to improve women's access to contraceptives address these gendered power inequalities by challenging gender norms and hegemonic masculinities of dominance.

However, it is also important to consider how programmes that aim to "empower" women and girls are cognizant of the danger of men feeling threatened by women's empowerment, and the possibility for gender-based violence and intimate partner violence as a reaction to this increased empowerment. Efforts to implement empowerment programming aimed at increasing women's self-efficacy for contraceptive use must consider the way in which sexual scripts and relationship dynamics, including unequal power, are likely to constrain women's ability to make decision [32]. Given the influence of socio-cultural level norms and hegemonic masculinities in sub-Saharan Africa, further attention needs to be paid to community level intervention approaches that aim to create conducive environments to enable women to exert their own agency and decide to contraceptive use [1, 33]. This can be achieved by shifting

problematic gender norms, restrictive masculinities and addressing gender inequalities that exist at the structural level [29].

Limited knowledge about contraceptive methods have been found as barrier for contraceptive use in general [34–36]. The population under study came from a community where sensitization on contraception was not a government priority. It is clear that when people do not know about a service or have limited knowledge about it, they would not seek to use the service [37]. The participants in this study reported that contraceptive information is not widely known or disseminated in their settlements. Additionally, they acknowledged that there was lack of health promotion messages back home in South Sudan that would encourage them to use contraceptives. Therefore, it is imperative to consider equipping refugees with information on contraception for better health outcomes.

At interpersonal level, the interaction with immediate context influenced the decisions to use contraceptives by refugee women. The other factors that relate to the environment around an individual that has affected contraceptive use were the key influencers such as clan leaders, mothers in law, spouses, peers, siblings, and service providers.

In this study, we established that gender dynamics played a key role in determining women's autonomy for decision making. In many patriarchal societies, the gender dynamics range from influence of health behavior to deciding on critical issues [38]. In most of these societies, men have dominated all decisions pertaining to family matters which concurs with a finding from a study done by Mejía-Guevara and others in India [39]. The male dominance has further exacerbated the vulnerability of women and girls affecting their access to health services. In some circumstance for women to access certain health services they must first get approval from their partners [40]. Due to the subordinate situation of the refugee women, they continuously faced challenges in accessing contraception. This study found that women could not make their own decision to use contraceptives because of male influence against the idea. The community leaders who are predominantly men are fully aware of this and are also enjoying the monopoly of making decisions on what type of healthcare should be accessed by women. This study concurs with the findings of a study done in Ethiopia [41] which revealed that men are regarded as the custodians of the community and have patriarchal dominance and responsibility of taking important decisions for their communities including contraceptive choices [42]. Drawing from the above findings, this study recommends a comprehensive tailored community programs to create awareness on the rights of every individual particularly right to health. While doing this, men may appreciate and pilot the process of re-considering the rules governing patriarchy while at the same time allow women to take part in decisions related to their health needs and choices.

Use of contraception is aimed at achieving pregnancy prevention. The finding of this study is similar with studies done by other scholars who confirmed that some communities perceived contraception as a characteristic of promiscuity [34, 43, 44]. The different cultural norms that have long viewed contraceptive use as synonymous to promiscuity have led to many refugee women fearing to access the services [34]. They added that the community viewed contraceptive use as a cover up for women's intentions to engage in sexual relationships with other men outside their marriage [44]. Given these concerns, there is need to clearly draw a line between pregnancy prevention particularly for child spacing and the community views on risk sexual behavior.

We also established that some participants viewed contraceptive use as a barrier to getting many cows in form of bride price that comes with girl children. This has had a subsequent effect on the aspiration of couples to jointly agree to have as many girl children as possible in anticipation for monetary value through marriage/dowry. This study further re-affirms the findings by XU Young and others who in their study revealed that commercialization of the girl child

through marriage and dowry payment is not new in many low- and middle-income countries [45]. The dowry issue is deep rooted in the traditions of the people of South Sudan. Families that produced many girls were assured of big gains in form of dowry. Thus, use of contraception is not guaranteed given the urge to produce many girls to reap the benefits of bridal price or dowry. Involving men in critical issues regarding decision making about contraception is crucial.

At community level, the broader view of contraception is equally significant in shaping one's decision about its use. Contraceptive use has for a long time been perceived differently by many communities across the world [46]. The various myths and misconceptions have impacted negatively on the use of contraception in many African countries [34, 35, 43]. Although many scholars and practitioners have written and talked about these myths, some refugees have continued to believe in them resulting in poor pregnancy outcomes [47]. In this study, the adolescent and young women expressed concern that their parents use these myths to scare them from using contraceptives, saying they will give birth to babies with missing body parts in future. This was confirmed by adult participants who also reported that contraceptive use led to children being born with deformities which has deterred some refugee women from making decision to use contraceptives. Owing to these findings above, there is need to raise awareness about contraception in the refugee settlements and dispel myths and misconceptions.

Cultural norms and values have a direct impact on the uptake of health services which may positively or negatively affect the health outcomes of the people [48]. The finding of this study is similar with a finding in a study done by Kapadia-Kundu and others [41] which revealed that many communities across Africa view women who use contraceptives as weak and fear to raise children, as perceived to be against the community expectation and the cultural norms. Women were expected to give birth until menopause and to have a big family. Due to these strong cultural norms, women who fail to have the desired number of children are subjected to disciplinary action like being chased out of the marital home or face intimate partner violence [29]. Considering this finding, we recommend that the Ministry of Health and partners working in the refugee settlements roll out a health promotion campaign to enlighten community leaders and men on the selected cultural issues that impact on the decision to use contraceptives. Such an initiative can help in attitude change and increase the uptake of contraception among refugee communities.

Reprisal following contraceptive use is one of the ways in which men can restrict the uptake of certain health care services by women for their sexual reproductive health desires [35, 41]. This study further established that women who went ahead and used contraceptives without approval of their husbands and/or family members received some form of punishment including intimate partner violence or separation/divorce. A study done by Kapadia-Kundu and others in Ethiopia [41] revealed that such reprisals may involve neglecting women by leaving them without any social support thereby making them vulnerable considering their socio-economic status in the community. Most families have disintegrated due to disagreements on contraceptive use, impacting negatively on the lives of children and the women themselves due to intimate partner violence [49]. Thus, the persistent gender norms have affected uptake of health services amongst young women leading to fear [50]. The refugee women believed that use of contraceptives was safer when the husbands and family members were not aware. Considering the above fears, interpersonal communication should be designed for men, women, and local leaders to uphold the dignity of women and their rights to health. The men should also be sensitized about the consequences of intimate partner violence if they were acting out of ignorance. This will help them to appreciate the law and its implication on intimate partner violence while accepting contraceptive use by their women [51, 52].

Community leaders have always been viewed as gate keepers for any community programs. In this study, it was revealed that community leaders particularly the religious and other clan

leaders were antagonizing family planning programs within the settlements [53]. Their quest for many children to build a stronger society, have enough labour in the gardens and replace those who died during the war back home is openly demonstrated in their campaigns against contraception. This concurs with a finding from a study done in South Sudan which revealed that contraceptives are viewed as a barrier to achieving the desired number of children [53]. This is because they view children with another lens as source of prestige and security and failure to do so called for clan leaders' intervention and advocating for another wife.

At institutional level, this study explored some institutional factors that included health systems and not limited to availability of commodities and quality of contraceptive services that were crucial in influencing individual decision to use contraceptive. Participants reported that the health services around the refugee settlements were burdened with multiple challenges such as limited staff to offer quality contraceptive information and services in Adjumani and limited contraceptive methods available. This finding concurs with a study done by Anand and others who revealed that poor quality services and limited choice impacted negatively on the women's decision-making [7].

The existing hostile policy frameworks also affected certain age groups to access contraception. In Uganda, the current state of the Sexual Reproductive Health and Right policy guidelines restricts access to contraceptive information and services to adolescent girls below 16 years [54]. This has put the sexual reproductive health of adolescent girls in jeopardy, rendering them susceptible to teenage pregnancy.

## Implication for practice

There is need for a multidisciplinary approach to combatting barriers to contraception use among refugee communities. Professionals from different organizations and departments need to be educated and trained to understand the culture of refugee communities and effectively help in promoting access to contraception services. More importantly there is need for a robust health promotion programme targeting both men and women in the refugee communities. Such a health promotion initiative should also be supported and conducted by people within the refugee communities to enhance acceptability. More conversations are needed with refugee women and men to understand the best ways of caring for the women and tap into opportunities available to break the barriers to contraceptive use. Such engagement may be key to finding lasting solutions in ending opposition to contraceptive use among refugee communities. There is need to adopt locally appropriate approaches and interventions to deliver health promotion to contraceptive awareness and paying attention to concept of social justice and recognition of local knowledge and values.

## Limitation of the study

The research participants were drawn from one district in Uganda in this case Adjumani. Future research encompassing several districts hosting refugees in Uganda will be needed to enhance comparisons. Furthermore, the research study was limited to only a qualitative research paradigm. However, future studies using both qualitative and quantitative research paradigms will be needed to explore the research issues from different epistemological and ontological positions.

## Conclusion and recommendation

Several constrains negatively affected the use of contraception among South Sudanese refugee women living in Uganda. Gender dynamics were found to influence decisions for contraceptive use given the low empowerment of women and the Patriarchal dominance of men in

making decision for health choices. This research also found that cultural norms related to bride price and limited knowledge on the benefits of contraception affected its use. We also found that majority of the women were not engaged in economic activities. Hence were living in the mercy of their husbands for any decision about their sexual and reproductive choices.

We therefore recommended the following:

Ministry of Health and partners roll out health promotion awareness using community friendly strategies such as community dialogues to discuss benefits of family planning and clarity on myths and misconceptions, rights to health and upholding the dignity of women in society. In addition, acknowldegement of the values, knowledge and ethos of the refugee communities should be recognized to build a formidable relationship with the community leaders while promoting contraception at community level. More importantly, robust policies supporting contraceptive use among refugee communities are needed and the public must be aware of it for continued engagement with health services to improve women s' health. From the widely expressed views about the lack of financial capacity, we suggest that refugee women should be enrolled in financial literacy programs for socio—economic empowerment.

## Acknowledgments

We are grateful to all refugee women who took part in this research and the Research Assistants for data collection. Special thanks go to my supervisors for the continued support.

## Author Contributions

**Conceptualization:** Roselline Achola, Lynn Atuyambe, Elizabeth Nabiwemba, Christopher Garimoi Orach.

**Data curation:** Roselline Achola, Lynn Atuyambe, Mathew Nyashanu, Christopher Garimoi Orach.

**Formal analysis:** Roselline Achola, Lynn Atuyambe, Mathew Nyashanu, Christopher Garimoi Orach.

**Funding acquisition:** Roselline Achola, Lynn Atuyambe, Elizabeth Nabiwemba, Christopher Garimoi Orach.

**Investigation:** Roselline Achola, Mathew Nyashanu, Christopher Garimoi Orach.

**Methodology:** Roselline Achola, Lynn Atuyambe, Elizabeth Nabiwemba, Mathew Nyashanu, Christopher Garimoi Orach.

**Project administration:** Roselline Achola.

**Resources:** Roselline Achola, Lynn Atuyambe, Elizabeth Nabiwemba, Christopher Garimoi Orach.

**Software:** Roselline Achola, Mathew Nyashanu.

**Supervision:** Lynn Atuyambe, Elizabeth Nabiwemba, Christopher Garimoi Orach.

**Validation:** Roselline Achola, Lynn Atuyambe, Elizabeth Nabiwemba, Mathew Nyashanu, Christopher Garimoi Orach.

**Visualization:** Roselline Achola.

**Writing – original draft:** Roselline Achola, Mathew Nyashanu, Christopher Garimoi Orach.

**Writing – review & editing:** Roselline Achola, Lynn Atuyambe, Elizabeth Nabiwemba, Mathew Nyashanu, Christopher Garimoi Orach.

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
