## [Decision Letter · Decision Letter 0]

14 Mar 2023

PONE-D-22-31975Barriers to contraceptive use in humanitarian settings: Experiences of South Sudan refugee women living in Adjumani district, UgandaPLOS ONE

Dear Dr. Achola,

Thank you for submitting your manuscript to PLOS ONE. After careful consideration, we feel that it has merit but does not fully meet PLOS ONE’s publication criteria as it currently stands. Therefore, we invite you to submit a revised version of the manuscript that addresses the points raised during the review process.

We look forward to receiving your revised manuscript.

Kind regards,

Adetayo Olorunlana, Ph.D.

Academic Editor

PLOS ONE

5. Thank you for stating the following in the Acknowledgments/ Funding Section of your manuscript:

“Grant ID: NICHE -UGA-288. This research was conducted with financial support from Nuffic grant through The International Institute of Social Studies Erasmus University, Rotterdam for Strengthening Education and Training Capacity in Sexual Reproductive Health & Rights in Uganda (SET-SRHR).”

“Funding was only availed to RA, the Principal Investigator through Nuffic grant through The International Institute of Social Studies Erasmus University, Rotterdam for Strengthening Education and Training Capacity in Sexual Reproductive Health & Rights in Uganda (SET-SRHR). The funders had to role in the study design, data collection and analysis, decision to publish or preparation of the manuscript.”

6. We note that you have indicated that data from this study are available upon request. PLOS only allows data to be available upon request if there are legal or ethical restrictions on sharing data publicly. For more information on unacceptable data access restrictions, please see http://journals.plos.org/plosone/s/data-availability#loc-unacceptable-data-access-restrictions.

Reviewers' comments:

Reviewer's Responses to Questions

**Comments to the Author**

1. Is the manuscript technically sound, and do the data support the conclusions?

Reviewer #1: Yes

Reviewer #2: No

2. Has the statistical analysis been performed appropriately and rigorously? 

Reviewer #1: N/A

Reviewer #2: N/A

3. Have the authors made all data underlying the findings in their manuscript fully available?

Reviewer #1: Yes

Reviewer #2: Yes

4. Is the manuscript presented in an intelligible fashion and written in standard English?

Reviewer #1: Yes

Reviewer #2: No

5. Review Comments to the Author

Reviewer #1: The paper studied a unique and often disadvantage population, thus will make the work attract readership. But there are a few technical issues the authors need to address.

1. The authors did not present any responses from the age group 15-19, adolescents experiences of barriers surely differs from older women. It will add value to filter responses from such age group and reflect such.

2. The authors choice and use of word "the research participant" made the findings/result section boring. This word reflect in every theme. Kindly revise.

3. Discussion: The discussion section needs to be revised to allow for synergy in the barriers identified. Also the authors is advised to sieve all recommendations listed with this section to feed into the "conclusion and recommendation" section.

4. The authors should clearly cone down on how much of a barrier are the the intra personal level factors, inter personal level and community level. Are there specific demographics to be focused for programming. If data is available, how do the Refugee host community reflect their country of origin.

5. The study did not talk about availability of methods/ providers (Access to FP services with the host communities).

Reviewer #2: This manuscript explores barriers to contraceptive use in a humanitarian setting, specifically South Sudan refugees living in Uganda. The article makes a great contribution to our understanding of the barriers faced by women living in a humanitarian setting in accessing sexual and reproductive health services. However, several issues need to be addressed to improve the paper.

Overall comments

•To ensure that all necessary components of the study are reported, use the COREQ checklist

•The authors indicated the study was guided by the socioecological model; however, the analysis and results sections seem not to have been guided by the model as the key findings/themes are not mapped according to the various ecological levels.

Comments

•Several statements and data have been made without being referenced. Lines 62-64, 66-68, 69-70, 79-81, and 104-110 illustrate statements that should have been cited. Review the entire document to ensure all statements and data are correctly referenced.

•Review the statistics given from lines 104-108. No need to include the n-values.

•The introduction section is quite lengthy and does not flow. Restructure the content to ensure consistency and brevity in the flow of ideas

•Revise line 176-research assistance as opposed to research assistants.

•Repetition of the content listed under data collection and piloting the interview schedule. Did the study conduct both a pilot study and pretesting, if yes, what were their objectives?

•There is inadequate information about the study participants. How were the participants recruited? What were the inclusion/exclusion criteria? Who participated in IDIs versus FGDs? A table will be helpful to show the participants who were interviewed. Apart from age, were there any socio-demographic characteristics of the women collected to better understand who was interviewed?

•In the results section, the first quote used in the subtheme (Joint decision…) doesn’t fully fit. It illustrates contraceptive decisions being driven by men, and not jointly by the couple.

•Need to review when to use family planning and contraceptives.

•Line 362-Revise the title from discussions to discussion.

•The discussion section fails to compare and contrast previous studies done in similar settings.

•Paraphrasing the results without referencing other studies: Are these findings unique to

•Referencing and citation are poorly done across the document. Review the reference style both in the text and the reference list to ensure it conforms to the journal’s guidelines.

6. PLOS authors have the option to publish the peer review history of their article (what does this mean?). If published, this will include your full peer review and any attached files.

Reviewer #1: No

Reviewer #2: No

---

## [Author Response · Author response to Decision Letter 0]

28 May 2023

Response: The authors looked at the PLOS ONE template and adjusted the manuscript according (see attached revised manuscript).

Response: This section has been revised as follows: Written consent was obtained from participants. For the emancipated minors (15-17 years) who were married and had children, we obtained consent from them. However, for the other minors, we obtained assent and consent from their parents/guardians. (see ethical statement section)

Response: The PLOS LaTeX template has been used in the revised manuscript

4. We note that the grant information you provided in the ‘Funding Information’ and ‘Financial Disclosure’ sections do not match. When you resubmit, please ensure that you provide the correct grant numbers for the awards you received for your study in the ‘Funding Information’ section.

Response: This has been addressed: Grant ID: NICHE -UGA-288. The Nuffic grant through the International Institute of Social Studies Erasmus University, Rotterdam for Strengthening Education and Training Capacity in Sexual Reproductive Health & Rights in Uganda (SET-SRHR)

5. Thank you for stating the following in the Acknowledgments/Funding Section of your manuscript: “Grant ID: NICHE -UGA-288. This research was conducted with financial support from Nuffic grant through The International Institute of Social Studies Erasmus University, Rotterdam for Strengthening Education and Training Capacity in Sexual Reproductive Health & Rights in Uganda (SET-SRHR).”

“Funding was only availed to RA, the Principal Investigator through Nuffic grant through The International Institute of Social Studies Erasmus University, Rotterdam for Strengthening Education and Training Capacity in Sexual Reproductive Health & Rights in Uganda (SET-SRHR). The funders had to role in the study design, data collection and analysis, decision to publish or preparation of the manuscript.”

Response: The funding information has been declared in the funding statement as per the guidance. - Funding information has been removed from the Acknowledgment section and any other areas of our manuscript. The funding information has been included in the cover letter as guided. We acknowledge the support to change the online form on our behalf to include the amended funding statement as below. 

“The Principal Investigator received funding from Grant ID: NICHE - UGA-288. Nuffic grant through The International Institute of Social Studies Erasmus University, Rotterdam for Strengthening Education and Training Capacity in Sexual Reproductive Health & Rights in Uganda (SET-SRHR). 

The funders had no role in the study design, data collection and analysis and did not make the decision to publish or preparation of this manuscript”. 

6. We note that you have indicated that data from this study are available upon request. PLOS only allows data to be available upon request if there are legal or ethical restrictions on sharing data publicly. For more information on unacceptable data access restrictions, please see http://journals.plos.org/plosone/s/data-availability#loc-unacceptable-data-access-restrictions.

Response: This has been addressed in the cover letter as follows: There is no legal restriction for sharing de-identified data set. However, this is part of the bigger study and data can only be shared upon successful submission of the entire project. For now, we have relevant data included within the manuscript and supporting information.

Response: We have not uploaded the de-identified data set. We acknowledge your support to update for us the Data Availability statement on our behalf to reflect the information provided as above.

Reviewers' comments: Reviewer's Responses to Questions

Comments to the Author

1. Is the manuscript technically sound, and do the data support the conclusions?

Reviewer #1: Yes

Reviewer #2: No

2. Has the statistical analysis been performed appropriately and rigorously?

Reviewer #1: N/A

Reviewer #2: N/A

3. Have the authors made all data underlying the findings in their manuscript fully available?

Reviewer #1: Yes

Reviewer #2: Yes

4. Is the manuscript presented in an intelligible fashion and written in standard English?

Reviewer #1: Yes

Reviewer #2: No

5. Review Comments to the Author

Reviewer #1: The paper studied a unique and often disadvantage population, thus will make the work attract readership. But there are a few technical issues the authors need to address.

1. The authors did not present any responses from the age group 15-19, adolescents experiences of barriers surely differs from older women. It will add value to filter responses from such age group and reflect such.

Response: Thank you for this concern, indeed the adolescents experiences of barriers to contraceptive use are crucial. This has been included in the result with a quote and also in the discussion sections of the revised manuscript. For example "...Some parents say that adolescent girls may have babies with missing body parts. So, any abnormality is associated with use of FP. Even those who don’t produce after marriage are labelled as having used FP at early age. FP use among young people is very difficult in the settlements.

2. The authors choice and use of word "the research participant" made the findings/result section boring. This word reflect in every theme. Kindly revise.

Response: Yes, the monotony was noted and has been revised to avoid boring presentation of results section. Alternative words have been selected and used as in the revised manuscript.

3. Discussion: The discussion section needs to be revised to allow for synergy in the barriers identified. Also the authors is advised to sieve all recommendations listed with this section to feed into the "conclusion and recommendation" section. 

Response: The discussion section has been revised as per the revised manuscript. The recommendations that was listed in discussion section has also been filtered and taken to conclusion and recommendation section.

4. The authors should clearly cone down on how much of a barrier are the intra personal level factors, inter personal level and community level. Are there specific demographics to be focused for programming. If data is available, how do the Refugee host community reflect their country of origin.

Response: This has been addressed in line with the ecological model which clearly demonstrated the different factors at different levels. (see Fig 1. The Ecological model)

The specific demographics to be focused for programming are age and economic status. The adolescents whose access is affected by the cultural norms of early marriage and the need for many children as they are married off at young age. Due to poverty, women remain dependent on men and obey what they dictate to continue getting social support for their family social needs.

Given the available data, the Refugee host community are privileged with various sources of information and the government programs that promote family planning. Thus, have better options in their own country. However, because of the above factors, they still have face barriers at different levels that affect their decision to use contraceptives.

5. The study did not talk about availability of methods/ providers (Access to FP services with the host communities).

Answer: Data on the available methods of family planning have not been considered in the revised manuscript. However, the most preferred method among the refugee community was condoms, whose access was easy within the settlements through health facilities and the community health workers. Other methods were also available like injectables and pills. Therefore, the women did not mention any significant barrier related to method availability but just the other cultural norms that affected the way decisions are made to access these methods.

Reviewer #2: This manuscript explores barriers to contraceptive use in a humanitarian setting, specifically South Sudan refugees living in Uganda. The article makes a great contribution to our understanding of the barriers faced by women living in a humanitarian setting in accessing sexual and reproductive health services. However, several issues need to be addressed to improve the paper.

Overall comments

•To ensure that all necessary components of the study are reported, use the COREQ checklist

Response: Yes, the consolidated criteria for reporting qualitative research (COREQ checklist) (Tong Alison, 2007) has been used to report the relevant components of the study as in the revised manuscript. 

•The authors indicated the study was guided by the socioecological model; however, the analysis and results sections seem not to have been guided by the model as the key findings/themes are not mapped according to the various ecological levels. 

Response: The analysis and result sections of this study has been revised and presented in line with the socioecological model. For example in the analysis, the participants involved in this study were women above 15-49 years of age. A total of 32 participants were involved in FGDs and fourteen women reached with IDIs). To avoid repetition, we have sign posted readers to the conceptual framework which shows the results. Some factors were found to be at more than one level.

The result section has been guided by the model as follows:

 1) At intra personal level, the factors that affected contraceptive use were, demographic and social factors. These included age, education, gender dynamics, limited knowledge, lack of partner support due to negative attitudes on contraception.

2) At the inter personal level, the factors were due to significant others which included Husbands/men’s negative attitudes, service providers, community/clan leaders antagonism on contraception, 

3) At community level, the factors included cultural norms, socio economic factors, antagonism by Religious leaders, misconceptions, Gender dynamics. 

Comments

•Several statements and data have been made without being referenced. Lines 62-64, 66-68, 69-70, 79-81, and 104-110 illustrate statements that should have been cited. 

Review the entire document to ensure all statements and data are correctly referenced.

•Review the statistics given from lines 104-108. No need to include the n-values.

Response: Yes, the authors have taken note and made revisions as in the revised manuscript. Proper citation has also been done. 

•The introduction section is quite lengthy and does not flow. Restructure the content to ensure consistency and brevity in the flow of ideas.

Response: The introduction section has been revised accordingly. 

•Revise line 176-research assistance as opposed to research assistants.

Response: The authors took note of the error and changed to research assistants in the revised manuscript 

•Repetition of the content listed under data collection and piloting the interview schedule. Did the study conduct both a pilot study and pretesting, if yes, what were their objectives?

Response: Only the pre-test was conducted. The objective was to check clarity of questions and the understanding by participants. Secondly, it was to ensure familiarity of the tool by the research assistants.

•There is inadequate information about the study participants. How were the participants recruited? What were the inclusion/exclusion criteria? Who participated in IDIs versus FGDs? A table will be helpful to show the participants who were interviewed. Apart from age, were there any socio-demographic characteristics of the women collected to better understand who was interviewed?

Response: the authors have revised this section and included a table of study participants, brief on inclusion/exclusion criteria and other socio-demographic characteristics of the participants as per the revised manuscript.

•In the results section, the first quote used in the subtheme (Joint decision…) doesn’t fully fit. It illustrates contraceptive decisions being driven by men, and not jointly by the couple.

Response: Although the contraceptive decision is driven by men, the end result is that they jointly agree to have many children and thus avoid contraceptive use. This is different from some women who do not agree to the suggestions by the partner and still go ahead to use contraceptives in hiding. Therefore, the authors have not changed the quote basing on that argument.

•Need to review when to use family planning and contraceptives.

Response: The authors have taken note. Family planning is used interchangeably with contraceptives given the circumstance and when referring to methods

•Line 362-Revise the title from discussions to discussion.

•The discussion section fails to compare and contrast previous studies done in similar settings.

Response: The authors have taken note of the comments on the discussion section and considered comparing and contrasting previous studies.

•Paraphrasing the results without referencing other studies: Are these findings unique to

•Referencing and citation are poorly done across the document. Review the reference style both in the text and the reference list to ensure it conforms to the journal’s guidelines.

Response: We have addressed the concerns as per the revised manuscript.

---

## [Decision Letter · Decision Letter 1]

16 Aug 2023

PONE-D-22-31975R1Barriers to contraceptive use in humanitarian setting: Experiences of South Sudan refugee women living in Adjumani district, Uganda; An explaratory qualitative studyPLOS ONE

Dear Dr. Achola,

Thank you for submitting your manuscript to PLOS ONE. After careful consideration, we feel that it has merit but does not fully meet PLOS ONE’s publication criteria as it currently stands. Therefore, we invite you to submit a revised version of the manuscript that addresses the points raised during the review process.

We look forward to receiving your revised manuscript.

Kind regards,

Adetayo Olorunlana, Ph.D.

Academic Editor

PLOS ONE

Journal Requirements:

Reviewers' comments:

Reviewer's Responses to Questions

**Comments to the Author**

1. If the authors have adequately addressed your comments raised in a previous round of review and you feel that this manuscript is now acceptable for publication, you may indicate that here to bypass the “Comments to the Author” section, enter your conflict of interest statement in the “Confidential to Editor” section, and submit your "Accept" recommendation.

Reviewer #3: (No Response)

Reviewer #4: All comments have been addressed

2. Is the manuscript technically sound, and do the data support the conclusions?

Reviewer #3: Yes

Reviewer #4: Yes

3. Has the statistical analysis been performed appropriately and rigorously? 

Reviewer #3: N/A

Reviewer #4: N/A

4. Have the authors made all data underlying the findings in their manuscript fully available?

Reviewer #3: Yes

Reviewer #4: No

5. Is the manuscript presented in an intelligible fashion and written in standard English?

Reviewer #3: Yes

Reviewer #4: Yes

6. Review Comments to the Author

Reviewer #3: Whilst this manuscript addresses an important issue, and presents interesting data, overall there are many typos and grammatical errors, which distract from the content. The entire manuscript would benefit from a close copy edit. See below for specific comments on errors picked up. I have also made a few general comments for the authors to consider, as well as suggestions for further reading which may help to enhance the discussion.

- Title on Cover page has spelling error

- Line 35: contraceptive use is life saving? or contraceptives can be life saving?

- Lines 37-38: I would avoid the term ‘significant other’ in academic writing – what does it really mean? A more objective precise term would be sexual/ romantic partner.

- Line 46: “women of similar categories” – do you mean participants were as homogenous as possible, with similar characteristics in terms of age, gender etc?

- Line 70: Could remove the ‘And’ at the start of the sentence.

- Line 75: typo: pillar

- Line 88: barriers plural?

- Line 90: Could remove the ‘And’ at the start of the sentence – just star the sentence with “yet”

- Line 100: how are they defined as being ‘vulnerable’ ?

- Line 111: rather than ‘beating’ preferable to use a term like ‘intimate partner violence’

- Line 124: Rather than writing “Mcleroy added; I quote” – it is preferable in a manuscript to paraphrase the quote and reference it, rather than writing in the first person.

- Line 129: describes

- Line 143: “a purposive sampling approach”

- Line 148: poorly written - revise the sentence

- Line 162: were translated

- Line 178: “the next step”

- Line 234: “the influence of men”

- Line 236: “their husbands’ involvement”

- Line 237” check grammar and tenses of verbs

- Lines 253-254: words missing? Revise sentences

- Line 326: “institutional” does not need to be capitalised

- Lines 333-335: this information about the Sexual and Reproductive Health policy guidelines should be in the discussion section, not the findings

- Line 343: grammar errors

- Line 344: close the brackets

- Line 345: revise sentence

- Line 351: word missing – revise

- Line 353: sentence incomplete

- Line 354: “take blood samples”

- Line 384: “campaign against” – not de-campaign

- Line 413: grammar errors – revise

- Line 416: revise – remove reference to the researcher – rather refer to the authors, or the study

- Line 436: decision making?

- Line 455: grammar errors

In the discussion, the authors refer to “empowerment”. Whilst the concept of “empowerment” has been central to efforts to address gendered inequalities and power imbalances, thereby accelerating development and improving the health and well-being of women and girls. Empowerment is framed as a process of enhancing an individual’s capacity to exercise choice, make decisions, and critically, the ability to act on those decisions, and achieving their choice. Interventions that aim to “empower” women and girls are designed to increase self-efficacy and agency, and therefore improve sexual and reproductive decision-making and related health outcomes – including the decision to use contraceptives or not. As this study demonstrates, since socio-cultural gendered norms and expectations are a key obstacle to women’s exercise of agency in choosing to use contraceptives or not – as pervasive gendered sexual norms tend to give men greater decision-making power in sexual and reproductive matters - it is critical that interventions and programmes that aim to improve women’s access to contraceptives address these gendered power inequalities by challenging gender norms and hegemonic masculinities of dominance. However, it also important to consider how programmes that aim to “empower” women and girls are cognisant of the danger of men feeling emasculated or threatened by women’s empowerment, and the possibility for gender based violence and intimate partner violence as a reaction to this increased empowerment. Efforts to implement empowerment programming aimed at increasing women’s self-efficacy for contraceptive use must consider the way in which dyadic level sexual scripts and relationship dynamics, including unequal power, are likely to constrain women’s enactment of their decision making. Given the influence of socio-cultural level norms and hegemonic masculinities in sub-Saharan Africa, further attention needs to be paid to community level intervention approaches that aim to create conducive environments that enable women to exert their own agency and decision making in contraceptive use by shifting problematic gender norms and restrictive masculinities, and addressing gender inequalities that exist at the structural level.

Prior research from the sub-Saharan African setting has indicated that men feel anxiety around the concept of female-controlled methods of contraception due to the belief that men should control fertility. There is a need to consider how giving women more agency in their sexual and reproductive health and choices, this may be perceived by men as a challenge to their control.

Suggested references :

- Karp C, Wood SN, Galadanci H, Sebina Kibira SP, Makumbi F, Omoluabi E, Shiferaw S, Seme A, Tsui A, Moreau C. (2020). ‘I am the master key that opens and locks’: Presentation and application of a conceptual framework for women’s and girls’ empowerment in reproductive health. Soc Sci Med, 258. https://doi. org/10.1016/j.socscimed.2020.113086.

- Robinson JL, Narasimhan M, Amin A, Morse S, Beres LK, Yeh PT, et al. Interventions to address unequal gender and power relations and improve self-efficacy and empowerment for sexual and reproductive health decision-making for women living with HIV: a systematic review. PLoS ONE. 2017;12(8):e0180699. https://doi.org/10.1371/journal.pone.0180699.

- Duby, Z., Bergh, K., Jonas, K. et al. “Men Rule… this is the Normal Thing. We Normalise it and it’s Wrong”: Gendered Power in Decision-Making Around Sex and Condom Use in Heterosexual Relationships Amongst Adolescents and Young People in South Africa. AIDS Behav (2022). https://doi.org/10.1007/s10461-022-03935-8

- Pool, R., G. Hart, G. Green, S. Harrison, S. Nyanzi, and J. Whitworth. 2000. “Men’s Attitudes to Condoms and Female Controlled Means of Protection against HIV and STDs in South-Western Uganda.” Culture, Health & Sexuality 2 (2): 197–211.

Reviewer #4: I see the authors have tried to address the previous reviewers comments. However, the use of COREQ is not clearly adhered to... there is no description of who conducted the interviews (gender, prior experience or relationship with the participants, and age cat of ther interveiwers...?). This information will strengthen the methodology followed in conducting and reporting this study...

the table of demographics is also not consistent and unclear why there is a column for % in the categories but none for the total column...? Also, there is use of decimal points sometimes and other times not...?

Otherwise, the paper is an important paper!

7. PLOS authors have the option to publish the peer review history of their article (what does this mean?). If published, this will include your full peer review and any attached files.

Reviewer #3: No

Reviewer #4: **Yes: **Kim Jonas, PhD

---

## [Author Response · Author response to Decision Letter 1]

27 Sep 2023

I take this opportunity to appreciate the reviewers for the very constructive comments given to us. We have reviewed the manuscript and responded accordingly as in the Manuscript.

---

## [Decision Letter · Decision Letter 2]

6 Nov 2023

PONE-D-22-31975R2Barriers to contraceptive use in humanitarian setting: Experiences of South Sudan refugee women living in Adjumani district, Uganda; An exploratory qualitative studyPLOS ONE

Dear Dr. Achola,

Thank you for submitting your manuscript to PLOS ONE. After careful consideration, we feel that it has merit but does not fully meet PLOS ONE’s publication criteria as it currently stands. Therefore, we invite you to submit a revised version of the manuscript that addresses the points raised during the review process.=

We look forward to receiving your revised manuscript.

Kind regards,

Adetayo Olorunlana, Ph.D.

Academic Editor

PLOS ONE

Journal Requirements:

Reviewers' comments:

Reviewer's Responses to Questions

**Comments to the Author**

1. If the authors have adequately addressed your comments raised in a previous round of review and you feel that this manuscript is now acceptable for publication, you may indicate that here to bypass the “Comments to the Author” section, enter your conflict of interest statement in the “Confidential to Editor” section, and submit your "Accept" recommendation.

Reviewer #3: (No Response)

2. Is the manuscript technically sound, and do the data support the conclusions?

Reviewer #3: Yes

3. Has the statistical analysis been performed appropriately and rigorously? 

Reviewer #3: N/A

4. Have the authors made all data underlying the findings in their manuscript fully available?

Reviewer #3: Yes

5. Is the manuscript presented in an intelligible fashion and written in standard English?

Reviewer #3: No

6. Review Comments to the Author

Reviewer #3: This revised version of the paper is much improved, however there are numerous grammatical and punctuation errors that need to be addressed. This paper would benefit from a close and thorough copy edit.

Line 97: injectables?

Line 104: “Women and girls are vulnerable because they need special care, support, and protection” ?? or “they are vulnerable and they need”… why do they need “special care, support, and protection” ?

Line 106: “makes them more susceptible” to what?

Lines 106-107: check punctuation

Line 125: using qualitative methods (should be methods not method)

Line 127: same comment re. methods

Line 129: describes

Lines 200-201: check punctuation – incomplete sentences

Lines 204-205: shouldn’t it be “South Sudanese refugee women”

Line 213: reached through FGDs? Makes it sound like this was the recruitment modality. Participated in?

Line 234: revise sentence - information on contraception is lacking

Line 267: “that personal decision” – which one? Sentence is unclear

Line 356: ovaries plural

Line 362: comma unnecessary

Line 409: sentence incomplete

Line 431: findings plural

Lines 444-445: sentence needs revision

Line 510: findings plural

Line 522: push on effect? Do you mean knock-on effect? Perhaps better to use more scientific /academic language – e.g. subsequent effect

Line 527: deep rooted

Line 549: sentence needs revision – grammar incorrect

Line 596: unfriendly policy framework? Hostile? What do you mean by unfriendly?

Line 626: dynamics were (dynamics are plural – check grammar)

Lines 631-632: check grammar and punctuation – sentence needs revision – comma is not in the right place

Line 634: sentence incomplete

7. PLOS authors have the option to publish the peer review history of their article (what does this mean?). If published, this will include your full peer review and any attached files.

Reviewer #3: No

---

## [Author Response · Author response to Decision Letter 2]

5 Dec 2023

I thank you so much for taking time to review this manuscript and the valuable comments given in regards to this paper. I have responded to all the comments as in the response letter.

---

## [Decision Letter · Decision Letter 3]

27 Dec 2023

Barriers to contraceptive use in humanitarian setting: Experiences of South Sudanese refugee women living in Adjumani district, Uganda; An exploratory qualitative study

PONE-D-22-31975R3

Dear Dr. Achola,

We’re pleased to inform you that your manuscript has been judged scientifically suitable for publication and will be formally accepted for publication once it meets all outstanding technical requirements.

Kind regards,

Adetayo Olorunlana, Ph.D.

Academic Editor

PLOS ONE

Additional Editor Comments (optional):

Reviewers' comments:

Reviewer's Responses to Questions

**Comments to the Author**

1. If the authors have adequately addressed your comments raised in a previous round of review and you feel that this manuscript is now acceptable for publication, you may indicate that here to bypass the “Comments to the Author” section, enter your conflict of interest statement in the “Confidential to Editor” section, and submit your "Accept" recommendation.

Reviewer #3: All comments have been addressed

2. Is the manuscript technically sound, and do the data support the conclusions?

Reviewer #3: Yes

3. Has the statistical analysis been performed appropriately and rigorously? 

Reviewer #3: N/A

4. Have the authors made all data underlying the findings in their manuscript fully available?

Reviewer #3: Yes

5. Is the manuscript presented in an intelligible fashion and written in standard English?

Reviewer #3: Yes

6. Review Comments to the Author

Reviewer #3: (No Response)

7. PLOS authors have the option to publish the peer review history of their article (what does this mean?). If published, this will include your full peer review and any attached files.

Reviewer #3: No

---

## [Editor Report · Acceptance letter]

21 Feb 2024

PONE-D-22-31975R3 

PLOS ONE

Dear Dr. Achola, 

I'm pleased to inform you that your manuscript has been deemed suitable for publication in PLOS ONE. Congratulations! Your manuscript is now being handed over to our production team.

Kind regards, 

on behalf of

Associate Professor Adetayo Olorunlana 

Academic Editor

PLOS ONE